# The Prognostic-Based Approach in Growth Hormone-Secreting Pituitary Neuroendocrine Tumors (PitNET): Tertiary Reference Center, Single Senior Surgeon, and Long-Term Follow-Up

**DOI:** 10.3390/cancers15010267

**Published:** 2022-12-30

**Authors:** Abel Ferrés, Luís Reyes, Alberto Di Somma, Thomaz Topczewski, Alejandra Mosteiro, Giulia Guizzardi, Andrea De Rosa, Irene Halperin, Felicia Hanzu, Mireia Mora, Isam Alobid, Iban Aldecoa, Núria Bargalló, Joaquim Enseñat

**Affiliations:** 1Department of Neurosurgery, Hospital Clínic de Barcelona, 08036 Barcelona, Spain; 2Faculty of Medicine, University of Barcelona, 08036 Barcelona, Spain; 3Department of Neurosurgery, Policlinico Federico II, 80131 Napoli, Italy; 4Department of Endocrinology, Hospital Clínic de Barcelona, 08036 Barcelona, Spain; 5Department of Otorhinolaryngology, Hospital Clínic de Barcelona, 08036 Barcelona, Spain; 6Department of Anatomical Pathology, Hospital Clínic de Barcelona, 08036 Barcelona, Spain; 7Department of Radiology, Hospital Clínic de Barcelona, 08036 Barcelona, Spain

**Keywords:** acromegaly, pituitary gland, endoscopic endonasal approach, pituitary neuroendoendocrine tumor, long-term follow-up, prognostic factors

## Abstract

**Simple Summary:**

Pituitary neuroendocrine tumors (PitNET) are characterized to present a heterogeneous behavior, and growth hormone (GH)-secreting PitNET is not an exception. Promptly determining which patients are affected by more aggressive tumors is essential to guide the optimal postoperative decision-making process [prognostic-based approach]. In this study, the authors determined younger age, higher preoperative GH and- or IGF-1 levels, group 2b of the clinicopathological classification, Knosp’s grade IV, MRI T2-weighted tumor hyperintensity, and sparsely granulated cytokeratin expression pattern are related to worse postoperative outcomes in long-term follow-up patients affected with GH-secreting PitNET.

**Abstract:**

Postoperative deserved outcomes in acromegalic patients are to normalize serum insulin-like growth factor (IGF-1), reduce the tumoral mass effect, improve systemic comorbidities, and reverse metabolic alterations. Pituitary neuroendocrine tumors (PitNET) are characterized to present a heterogeneous behavior, and growth hormone (GH)-secreting PitNET is not an exception. Promptly determining which patients are affected by more aggressive tumors is essential to guide the optimal postoperative decision-making process [prognostic-based approach]. From 2006 to 2019, 394 patients affected by PitNET were intervened via endoscopic endonasal transsphenoidal approach by the same senior surgeon. A total of 44 patients that met the criteria to be diagnosed as acromegalic and were followed up at least for 24 months (median of 66 months (26–156) were included in the present study. Multiple predictive variables [age, gender, preoperative GH and IGF-1 levels, maximal tumor diameter, Hardy’s and Knosp’s grade, MRI. T2-weighted tumor intensity, cytokeratin expression pattern, and clinicopathological classification] were evaluated through uni- and multivariate statistical analysis. Sparse probability of long-term remission was related to younger age, higher preoperative GH and- or IGF-1, group 2b of the clinicopathological classification, and sparsely granulated cytokeratin expression pattern. Augmented recurrence risk was related to elevated preoperative GH levels, tumor MRI T2-weighted hyperintensity, and sparsely granulated cytokeratin expression pattern. Finally, elevated risk for reintervention was related to group 2b of the clinicopathological classification, Knosp’s grade IV, and tumor MRI T2-weighted hyperintensity. In this study, the authors determined younger age, higher preoperative GH and- or IGF-1 levels, group 2b of the clinicopathological classification, Knosp’s grade IV, MRI T2-weighted tumor hyperintensity and sparsely granulated cytokeratin expression pattern are related to worse postoperative outcomes in long-term follow-up patients affected with GH-secreting PitNET.

## 1. Introduction

Acromegaly (ACM) due to growth hormone (GH) secreting pituitary neuroendocrine tumors (PitNET) is initially treated surgically if not contraindicated [1]. Therapeutic goals in patients with ACM are to lower the serum insulin-like growth factor-1 (IGF-1) concentrations within the normal range for the patient’s age and gender with or without suppressing GH (growth hormone) nadir to <1.0 ng/L following an oral load of 75 g glucose (oral glucose tolerance test: OGTT), control tumor size to reduce mass effects, improve systemic comorbidities and symptoms, and reverse metabolic alterations [2]. When IGF-1 concentrations return to typical, soft tissue overgrowth and related symptoms progressively reduce, and metabolic alterations such as diabetes mellitus improve, returning life expectancy to that of the general population [3,4].

Deleterious effects of elevated IGF-1 levels and heterogeneous behavior of pituitary neuroendocrine tumors (PitNET) in general, and GH-secreting PitNET in particular, make it indispensable to determine which prognostic factors are of significant utility. Several postoperative outcome predictors have been described in the literature as intended to predict progression, recurrence, and response to medical treatment [1]. It is noticed that younger age [5,6,7,8], female gender [7,9], elevated preoperative GH [3,6,8,9,10,11,12,13,14,15,16,17,18], augmented IGF-1 levels [8,9,12,14,19,20,21,22,23,24], higher Hardy’s grade [3,25,26], higher Knosp’s grade [5,8,9,11,13,14,21,24,27,28,29,30,31,32], higher maximal tumor diameter [9,15,29,33,34,35,36,37,38,39], tumor T2-hyperintensity [40,41,42,43,44], sparsely granulated cytokeratin expression pattern [45,46], elevated Ki-67 [6], and p53 expression [6,47] may be related with poor postsurgical results in terms of hormonal control, recurrence, and response to adjuvant medical treatment.

Special mention of the prognostic clinicopathological PitNET classification described by Trouillas et al. [48,49] must be made because it integrates multiple of the previously described predictive factors, thus facilitating the follow-up decision-making process.

The reduction in life expectancy associated with the heterogenic behavior inherent in PitNET makes an accurate prognostic-based approach in acromegalic patients indispensable. Identifying in which cases long-term disease control will be or not be accomplished after surgical treatment is paramount in defining the best follow-up decision-making process. Concerning the previous statement, there are few publications [27,50,51,52] evaluating the value of predictors in long-term follow-up (>24 months), making it difficult to determine in which subgroup of patients’ initial postsurgical results will be durable. 

Our study aims to determine which prognostic variables help predict long-term postoperative outcomes to guide in conceiving the most accurate management of acromegaly. Particular emphasis on evaluating the long-term predictive utility of the tumor T2 intensity and cytokeratin distribution pattern will be provided in this study due to the sparse literature on the subject. In addition, further validation of the prognostic value of the clinicopathological PitNET classification will be presented.

## 2. Materials and Methods

### 2.1. Study Population

The PitNET database of the Hospital Clínic, Barcelona, Spain, was queried to identify all patients with ACM operated via an endoscopic endonasal approach (EEA) between 2006 to 2019. From this database, constituted of 394 endoscopic resected PitNET, 44 patients affected by GH-secreting PitNET were identified and retrospectively included in the study. In all cases, the same senior surgeon performed the surgical procedure, and a strictly long-term follow-up of at least 24 months was accomplished. No medical treatment [somatostatin analogs] was administered before surgery. Patients with incomplete recorded clinical data, or subjects who received previous pituitary surgery, or radiotherapy, were excluded (*n* = 4). ACM was diagnosed using the standard criteria based on consensus guidelines: IGF-1 above the normal upper limit for age and gender and/or failure to suppress GH nadir to <1.0 ng/mL following OGTT [1], in association with the identification of the causative PitNET on MRI, and histological confirmation of the resected tumor. Remission criteria were also based on the same consensus guidelines and were defined as the last IGF-1 levels [measured at least 12 weeks after surgery] within the normal range for age and gender. GH nadir levels were also determined because these provide valuable information regarding normalizing the mortality risk when <1 μg/L [1]. In patients achieving remission without GH nadir control [<1 μg/L], GH nadir following OGTT was determined because the absence of GH nadir suppression [≤0.14 µg/L] is related to long-term remission [53]. Recurrence was defined as IGF-1 above the normal upper limit for age and gender once remission was previously achieved. The laboratory methods used to evaluate IGF-1 and GH levels were IMMULITE^®^ 20,000 IGF-1 and 2000/2500 GH, respectively [Siemens, Berlin, Germany]. Variables registered included demographic characteristics, tumor size, hormonal and metabolic parameters, PitNET classification, radiologic and histopathologic features, surgical complications, and follow-up time. This study involving human participants was reviewed and approved by Barcelona’s Clínic hospital ethical board. According to legislation, participant informed consent was unnecessary because of the study’s retrospective nature, anonymized recorded clinical data, and the impossibility of identifying participants directly or through identifiers in study results.

### 2.2. Preoperative Evaluation

The preoperative clinical and biochemical evaluations were cautiously performed employing standard methods. Hormonal profiles include basal GH, IGF-1, GH suppression after OGTT, and other pituitary axis hormones (PRL, TSH, and free T4; ACTH and cortisol, FSH/LH, and estradiol/testosterone). The radiologic evaluation was performed using magnetic resonance imaging (MRI). Sequences obtained were T1-weighted imaging (T1WI) spin-echo (SE) with and without contrast, dynamic contrast MRI T1WI fast turbo spin-echo (FSE), T2-weighted imaging (T2WI), and diffusion-weighted imaging (DWI). An expert neuroradiologist evaluated all imaging studies and redacted a detailed report. Cavernous sinus invasion was stratified using Knosp’s grading system [54], sphenoid sinus invasion and suprasellar extension were stratified using Hardy’s grading system [55], and T2 intensity of GH PitNET were visually classified as follows: tumors appearing equally or less intense than temporal lobe’s white matter were classified as hypointense, tumors appearing the same or brighter than temporal lobe’s gray matter were classified as hyperintense, and tumors’ intensity between both previous described were classified as isointense [43] (Figure 1).

### 2.3. Surgical Technique

MRI-based optic stereotactic navigation (Stealth Station, Medtronic, Inc., Minneapolis, MI, USA) was used in selected cases. The procedure was visualized with a 4 mm 0° endoscope (Karl Storz Endoscopy, Tuttlingen, Germany) attached to a high-definition (1080p) endoscopy camera (A3, Karl Storz Endoscopy) and monitor. A binostril approach was used in every case. The principal senior surgeon holds the camera during the procedure and is assisted by a second surgeon. If necessary, thirty-degree scopes were utilized at the end of the procedure to inspect the cavernous walls and suprasellar cistern for residual tumor [56,57,58,59,60,61,62].

### 2.4. Postoperative Evaluation

Steroids were routinely administered postoperatively, and diuresis was strictly quantified. Desmopressin was administered if necessary. Patients were discharged on the second day postoperatively if there were no complications. IGF-1 and GH nadir levels were performed 3–6 months after surgery. GH nadir suppression after OGTT was realized in patients who achieved recurrence [normal IGF-1 levels] without GH nadir levels normalization. After that, for patients in remission [normal IGF-1 levels], IGF-1 was assessed yearly. IGF-1 was checked at 3 month intervals for patients not in remission until it was achieved spontaneously or with medical therapy. Imaging follow-up evaluation using MRI was obtained 3 months after surgery in all cases. After that, a new MRI was performed to determine if the best treatment would be surgical, medical, or combined in case of recurrence of suspicion or insufficient response to the treatment. The resection rate was calculated using a volumetric comparison between the initial and follow-up MRI through a 1.5 T scanner (Siemens) through a semiautomatic region of interest (ROI) analysis with Iplan cranial v.3.0 software (Brainlab^®^, Feldkirchen, Germany).

A minimum postoperative follow-up of 24 months was accomplished in all patients included in the study.

As part of the anatomopathological evaluation of the specimen, the cytokeratin staining pattern was studied to differentiate between sparsely and densely granulated PitNET. All analyses were performed by an expert neuropathologist using anti-CAM 5.2 antibodies. GH-secreting PitNETs were classified as sparsely granulated if a cytokeratin dot pattern (fibrous bodies) was identified and densely granulated if a perinuclear pattern was detected. Fibrous bodies may be present in a percentage of densely granulated GH-secreting PitNET. In such cases, the absence of cadherin-E permits the classification of the specimen in the sparsely granulated group [63,64,65,66]. Nowadays, no transitional/intermediate forms are contemplated.

Finally, patients were classified into five groups depending on the tumor’s invasiveness and proliferation using PitNET clinicopathological classification described by Trouillas et al. [48,49]: Invasion was defined as histological and/or radiological (using magnetic resonance imaging; MRI) signs of cavernous or sphenoid sinus invasion. Proliferation was considered in the presence of at least two of three of the following criteria: Ki67 >1% (Bouin-Hollande fixative) or ≥ 3% (formalin fixative), mitoses > 2/10 high-power field, and- or P53 positivity (>10 strongly positive nuclei/10 high-power field). Group 1 corresponds to non-invasive tumors (1a: non-invasive and non-proliferative, 1b: non-invasive and proliferative), group 2 corresponds to invasive tumors (2a: invasive and non-proliferative, 2b: invasive and proliferative), and group 3 corresponds to pituitary carcinomas.

### 2.5. Statistical Analysis

A descriptive analysis was initially performed and presented using absolute and relative frequencies for categoric variables and median and range for continuous variables. IGF-1 levels have been included in the statistical analyses using the percentage of the normal upper limit of the age-adjusted normal range (%ULN IGF-1). Univariate analysis was calculated using the chi-square test, and multivariate analysis was calculated using logistic regression. Results from the univariate analysis are presented using Relative Risk (RR), its confidence interval, and statistical significance (*p*-value). Results from the multivariate analysis are presented through Relative Risk (RR) and statistical significance (*p*-value). All data were analyzed using IBM SPSS Statistics v.25.0 for Windows.

## 3. Results

From 2006 to 2019, 394 patients affected by PitNET were intervened via endoscopic endonasal transsphenoidal approach by the same senior surgeon. A total of 44 patients that met the criteria to be diagnosed as acromegalic and were followed up at least for 24 months (median of 66 months (26–156) were included in the present study (Figure 2). 

The participants’ median age at diagnosis was 50 (30–80) and participants were more frequently men (26; 59.1%). Tumors were more commonly macroadenomas (≥10 mm of maximum diameter) with or without minimal cavernous sinus invasion; Knosp’s grade 0 or I (17; 38.6% and 10; 22.7%, respectively), and without suprasellar expansion or sellar floor invasion. Lesions were principally iso- or hyperintense (21; 47.7% and 19; 43.2%, respectively) in T2-weighted M.R.I., slightly more frequently densely granulated when evaluating cytokeratin expression pattern (24; 54.5%) and primarily categorized in groups 1a and 2a (26; 59.1 and 12; 27.3%, respectively) in the clinicopathological classification. The resection rate was more than 90% of the lesion in 42 (95.4%) cases, and, from them, gross total resection (>99%) was achieved in 29 (65.9%) cases. The long-term postoperative remission was accomplished in 31 (70.5%) patients. Adjuvant therapy with somatostatin receptor analogs was administrated in 8 (18.2%) cases, 13 (29.5%) patients presented recurrence after surgery, and 5 (11.4%) were reoperated. The summary of the patient’s characteristics is presented in Table 1.

Initially, a **univariate** (Table 2 and Table 3) analysis is presented. The PitNET clinicopathological classification, cytokeratin expression pattern, and intensity in T2-weighted M.R.I. were analyzed for long-term IGF-1 normalization (long-term postoperative remission), recurrence, and need for reintervention.

First, the authors evaluated the predictive applicability of the clinicopathological classification. Group 2a was related to a diminished probability for long-term IGF-1 normalization **(RR = 0.1333 (0.092–0.629), *p* = 0.45)**. In addition, group 2b was associated with a low probability of achieving long-term IGF-1 normalization **(RR = 0.065 (0.006–0.660), *p* = 0.015)**, and significant recurrence risk **(RR = 4.875 (2.628–9.042), *p* = 0.001)**.

Regarding MRI and T2-weighted intensity, hyperintense lesions were related to a reduced probability for IGF-1 normalization **(RR = 0.091 (0.017–0.494), *p* = 0.045)**, elevated risk for recurrence **(RR = 13.444 (2.470–73.192), *p* = 0.001)**, and a significant need for reintervention **(RR = 1.941 (1.291–12.950) *p* = 0.046**.

Finally, when analyzing cytokeratin expression pattern, sparsely granulated was related to a reduced probability for IGF-1 normalization **(RR = 0.036 (0.004–0.317), *p* = 0.01)**, augmented recurrence risk **(RR = 34.5 (3.850–350.158), *p* = 0.01)**, and significant need for reintervention **(RR = 1.333 (1.035–1.717), *p* = 0.014)**. On the contrary, the densely granulated pattern was related to an augmented probability for IGF-1 normalization **(RR = 28.111 (3.154–250.516), *p* = 0.01)**, a minor recurrence risk **(RR = 0.029 (0.003–0.260), *p* = 0.01)**, and a reduced need for reintervention **(RR = 0.750 (0.582–0.966), *p* = 0.014)**.

Afterward, a **multivariate** analysis was realized, including the previously evaluated variables in addition to others widely evaluated in the literature: age, gender, preoperative GH and IGF-1 levels, maximal tumor diameter, Hardy’s grade, and Knosp’s grade. The results were as follows:

Younger age was associated with a low probability of achieving long-term IGF-1 normalization **(RR = 0.320 (0.125–0.780); (*p* = 0.034)**; higher preoperative GH levels were related to a minor chance of achieving IGF-1 normalization **(RR = 1.506 (1.205–4.606); (*p* = 0.045)** and increased recurrence risk **(RR = 1.700 (1.129–3.808); (*p* = 0.043)**; higher preoperative IGF-1 levels were associated with little probability of postoperative IGF-1 normalization **(RR = 1.490 (1.140–3.650); (*p* = 0.024)**; PitNET clinicopathological classification was related to sparse probability to achieve long-term remission and elevated need for reintervention in favor of group 2b **(RR = 0.080 (0.020–0.120); *p*= 0.028 and RR = 2.890 (2.120–4.460); *p* = 0.001, respectively)**; Knosp’s classification was related to a significant need for reintervention in favor of group IV **(RR = 4.360; (3.405–8.230); *p* = 0.011)**; hyperintense lesions in T2-weighted M.R.I. classification were associated with an augmented recurrence risk and reintervention **(RR = 8.704 (4.800–14.350); *p* = 0.026 and RR = 1.145 (1.032–2.445); *p* = 0.03, respectively)**; cytokeratin expression pattern was related to significant recurrence risk and sparse probability for long-term IGF-1 normalization for the sparsely granulated group **(RR = 10.433 (9.321–15.433); *p* = 0.002 and RR = 0.200 (0.045–0.776); *p* = 0.042, respectively)** (Table 4).

## 4. Discussion

Postoperative outcomes are variable among patients affected by GH-secreting PitNET because of the behavioral heterogenicity of the tumor. An exhaustive evaluation of predictive variables is paramount to deciding the best follow-up strategy (prognostic-based approach). Several prognostic factors have been widely analyzed in the literature, but in only a few studies has long-term follow-up of the participants been accomplished. Furthermore, variables such as tumor intensity in T2-weighted M.R.I. and cytokeratin expression pattern have been poorly studied for clinically relevant outcomes such as long-term remission, recurrence, and reintervention. 

In the present study, MRI. T2 hyperintense tumors were associated with a reduced probability for long-term remission, elevated recurrence risk, and a significant need for reintervention. From them, augmented recurrence risk and augmented demand for reintervention were consistently detected in the multivariate analysis. Our results are concordant with those published in the literature. It has been described that MRI-T2 hypointense tumors may present better short and long-term response profiles to first-line somatostatin analogs when compared with iso- or hyperintense lesions [40,42,43,67,68]. Although not yet implemented in our institution, immunohistochemical evaluation of the SSTR2 receptor may be useful to further recognize non-responders to first-line somatostatin analogs, especially in hyperintense T2-MRI GH-secreting PitNET intending to accurately guide postoperative medical treatment in case of necessity directly to second-line somatostatin analogs [pasireotide] [69]. Nowadays, high-performance monoclonal anti-SSTRs antibodies and well-established immunohistochemical protocols are available in the literature [69,70,71]. GH-secreting PitNET harboring moderate to strong membranous SSTR2 expression evaluated using an immunoreactive score (IRS) system with the cutoff punctuation ≥ 5 is reported to have a sensitivity of 86% and specificity of 91% in predicting hormonal control with first-line generation somatostatin analogs [71]. 

Sparsely granulated GH PitNET has been classically considered to present more aggressive behavior due to first-line somatostatin analogs resistance, increased invasiveness, and a tendency to early recurrence [55]. Our results are harmonious with those described in scientific publications. Sparsely-granulated GH PitNET was associated with a reduced probability of achieving long-term remission associated with augmented recurrence and reintervention risks. After multivariate analysis, the reduced probability of accomplishing long-term remission and elevated recurrence risk persisted as valuable predictive variables. On the contrary, densely-granulated GH PitNET presented antagonistic results. Previously predictive statements justified closer postoperative IGF-1 monitoring, precocious evaluation of SSTR2 expression if adjuvant medical treatment is needed, and prompt implementation of other postoperative therapeutic lines if necessary (pegvisomant, reintervention, radiotherapy, and temozolomide at last).

Because the clinicopathological classification proposed by Trouillas et al. [48,49] has been an essential contribution to the prognostic assessment of patients affected by PitNET, one of the author’s objectives was to contribute to further validation of it in this cohort of patients. It is an exciting classification as it integrates multiple predictive variables assessed independently in other publications. Its integration simplifies the decision-making process to achieve the best prognostic-based approach for treating patients affected by GH-secreting PitNET.

Our results went in the same direction as those previously presented in the original article. Especially proliferation, more than only invasion, plays an essential role in GH PitNET aggressiveness, which is traduced in a low probability of achieving long-term remission and higher recurrence risk. This aggressive profile was objectively determined for group 2. Specifically, groups 2a and 2b showed a reduced chance of achieving long-term postoperative remission, and in addition, group 2b presented an elevated recurrence risk. After multivariate analysis, the same prognostic value was maintained for group 2b but not for group 2a. Close follow-up is highly recommended for patients in group 2, particularly for those in group 2b. It is not anecdotic the necessity for radiation therapy because of failure of reintervention and adjuvant medical treatment [72]. Furthermore, there is a high risk of failure of all previously adjuvant therapies mentioned, thus resulting in the necessity to initiate temozolomide [66]

To obtain a complete evaluation of the usefulness of the prognostic factors in acromegaly, those classically studied in the literature were included in the multivariate analysis. Among them, especially younger individuals with elevated preoperative GH or IGF-1 levels and extensive cavernous sinus invasion may present worse postoperative outcomes, thus implying closer long-term follow-up.

Optimizing the prognostic-based approach for patients affected by GH-secreting PitNET requires an accurate evaluation of the utility of different predictive variables that permit prompt application of the appropriate postoperative follow-up strategy to control this chronically progressive disease and return our patient’s life expectancy and quality to that of the general population.

### Study Limitations

The retrospective nature of the study may limit bias control. The relatively small sample size due to the disease’s low prevalence and incidence may challenge the finding of subtle differences, especially between groups of the clinicopathological classification. Despite limitations, univariate and multivariate analyses of the different prognostic variables were correctly carried out.

## 5. Conclusions

Younger age, higher preoperative GH and- or IGF-1 levels, group 2b of the clinicopathological classification, Knosp’s grade IV, MRI. T2-weighted tumor hyperintensity and sparsely granulated cytokeratin expression pattern are related to worse postoperative outcomes in long-term follow-up patients affected with GH-secreting PitNET.

## Figures and Tables

**Figure 1 cancers-15-00267-f001:**
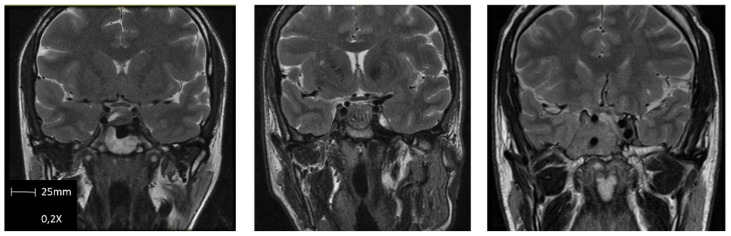
Examples of GH PitNET T2-weighted M.R.I. intensity classification. From **left** to **right**: Hypointense, isointense, and hyperintense.

**Figure 2 cancers-15-00267-f002:**
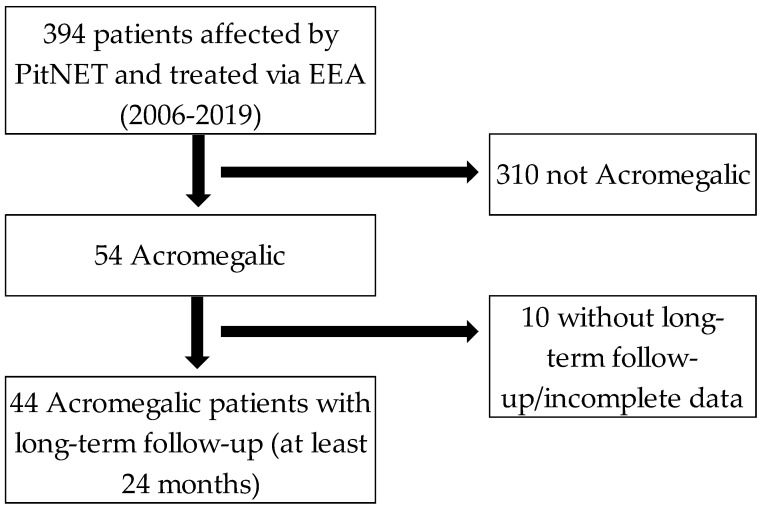
Inclusion criteria flow-chart. PitNET: Pituitary neuroendocrine tumors. EEA: Endoscopic endonasal approach.

**Table 1 cancers-15-00267-t001:** Patient’s characteristics.

Characteristics	All Patients (*n* = 44)
Age (years)	
Median	50
Range	30–80
Gender *n*, (%)	
Male	26 (59.1)
Female	18 (40.9)
Tumor size *n*, (%)	
Microadenoma (<10 mm)	10 (22.7)
Macroadenoma (≥10 mm)	34 (77.3)
Hardy’s classification	
Sellar invasion *n*, (%)	
0	0
I	7 (15.9)
II	26 (59.1)
III	3 (6.8)
IV	8 (18.2)
Suprasellar/cavernous sinus invasion *n*, (%)	
No	25 (56.8)
A	8 (18.2)
B	6 (13.6)
C	0
D	0
E	5 (11.4)
Knosp’s classification *n*, (%)	
0	17 (38.6)
I	10 (22.7)
II	6 (13.6)
IIIA	3 (6.8)
IIIB	2 (4.5)
IV	5 (11.4)
Preoperative GH levels (ng/mL)	
Median	4.18
Range	0.8–54
Preoperative IGF-1 levels (ng/mL)	
Median	701
Range	200–1456
%ULN IGF-1	290.56
T2-weighted M.R.I. intensity *n*, (%)	
Hypointense	4 (9.1)
Isointense	21 (47.7)
Hyperintense	19 (43.2)
Cytokeratin expression pattern *n* (%)	
Densely granulated	24 (54.5)
Sparsely granulated	20 (45.5)
Ki-67 (%)	
Median	1
Range	1–6
p53 expression *n* (%)	5 (10.9)
Mitoses (number)	
Median	1
Range	1–6
Preoperative campimetry *n*, (%)	
Normal	40 (90.9)
Bitemporal hemianopsia	3 (6.8)
PitNET classification *n*, (%)	
1a	26 (59.1)
1b	1 (2.3)
2a	12 (27.3)
2b	5 (11.4)
3	0
Hospital length of stay (days)	
Median	2
Range	1–15
Resection rate *n*, (%)	
Gross total resection (>99%)	29 (65.9)
Subtotal resection (90–99%)	13 (29.5)
Partial resection (<90%)	2 (4.5)
Long-term remission (IGF-1 normalization) *n*, (%)	
Yes	31 (70.5)
No	13 (29.5)
Adjuvant therapy (first-line somatostatin receptor analogs) *n*, (%)	
Yes	8 (18.2)
No	36 (81.2)
Recurrence *n*, (%)	
Yes	13 (29.5)
No	31 (70.5)
Need for reintervention *n*, (%)	
Yes	5 (11.4)
No	39 (88.6)
Follow-up campimetry *n*, (%)	
Improvement	1 (2.3%)
Unchanged	43 (99.7%)
Follow-up time (months)	
Median	66
Range	26–156

Ki-67 expression was evaluated in formalin fixative. Positivity for p53 expression was considered if the specimen presented >10 strongly positive nuclei/10 high-power field. The number of mitoses was evaluated in the high-power field. Cavernous sinus invasion was determined either if founded histologically and/or radiologically.

**Table 2 cancers-15-00267-t002:** Univariate analysis is presented. Each subgroup of the predictive variable clinicopathological classification is presented according to its statistically significant relation to the outcome variables (remission, recurrence, and reintervention) through Relative Risk and *p*-Value.

Predictive Variable	Subgroup	Outcome Variable	Relative Risk (RR)	*p*-Value
**Clinicopathological** **classification**	1a	Long-term remission	2.67 (0.874–1.999)	0.137
		Recurrence	0.298 (0.077–1.145)	0.072
		Reintervention	0.417 (0.062–2.791)	0.325
	1b	Long-term remission	1.387 (1.152–1.671)	0.427
		Recurrence	1.433 (1.177–1.745)	1
		Reintervention	1.132 (1.015–1.261)	1
	2a	Long-term remission	**0.1333 (0.092–0.629)**	**0.45**
		Recurrence	0.733 (0.163–3.304)	0.733
		Reintervention	1.933 (0.281-13.295)	0.417
	2b	Long-term remission	**0.065 (0.006–0.660)**	**0.015**
		Recurrence	**4.875 (2.628–9.042)**	**0.001**
		Reintervention	2.188 (0.194–24.679)	0.47

**Table 3 cancers-15-00267-t003:** Univariate analysis is presented. Each subgroup of the predictive variables analyzed is presented according to its statistically significant relation to the outcome variables (remission, recurrence, and reintervention) through Relative Risk and *p*-Value.

Predictive Variable	Subgroup	Outcome Variable	Relative Risk (RR)	*p*-Value
**MRI T2-weighted** **intensity**	Hyperintensity	Long-term remission	**0.091 (0.017–0.494)**	**0.045**
		Recurrence	**13.444 (2.470–73.192)**	**0.001**
		Reintervention	**1.941 (1.291–12.950)**	**0.046**
**Cytokeratin expression pattern**	Sparsely granulated	Long-term remission	**0.036 (0.004–0.317)**	**0.01**
		Recurrence	**34.5 (3.850–350.158)**	**0.01**
		Reintervention	**1.333 (1.035–1.717)**	**0.014**
	Densely granulated	Long-term remission	**28.111 (3.154–250.516)**	**0.01**
		Recurrence	**0.029 (0.003–0.260)**	**0.01**
		Reintervention	**0.750 (0.582–0.966)**	**0.014**

**Table 4 cancers-15-00267-t004:** Multivariate analysis is presented. Each subgroup of the predictive variable analyzed is presented according to its statistically significant relation to the outcome variables (remission, recurrence, and reintervention) through Relative Risk and *p*-Value.

Predictive Variable	Subgroup	Outcome Variable	Relative Risk (RR)	*p*-Value
**AGE**	Young	Long-term remission	**0.320 (0.125–0.780)**	**0.034**
**Preoperative GH**	Elevated	Long-term remission	**1.506 (1.205–4.606)**	**0.045**
		Recurrence	**1.700 (1.129–3.808)**	**0.043**
**Preoperative IGF-1**	Elevated	Long-term remission	**1.490 (1.140–3.650)**	**0.024**
**Clinicopathological classification**	2b	Long-term remission	**0.080 (0.020–0.120)**	**0.028**
		Reintervention	**2.890 (2.120–4.460)**	**0.001**
**Knosp’s classification**	IV	Reintervention	**4.360; (3.405–8.230)**	**0.011**
**MRI T2-weighted intensity**	Hyperintensity	Recurrence	**8.704 (4.800–14.350)**	**0.026**
		Reintervention	**1.145 (1.032–2.445)**	**0.03**
**Cytokeratin expression pattern**	Sparsely granulated	Long-term remission	**0.200 (0.045–0.776)**	**0.042**
		Recurrence	**10.433 (9.321–15.433)**	**0.002**

## Data Availability

The data presented in this study are available on request from the corresponding author.

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
