# Peer review of "The Prognostic-Based Approach in Growth Hormone-Secreting Pituitary Neuroendocrine Tumors (PitNET): Tertiary Reference Center, Single Senior Surgeon, and Long-Term Follow-Up"

_cancers, 2022, doi:10.3390/cancers15010267_

Round 1
Reviewer 1 Report
The Authors retrospectively analyzed data from 44 acromegalic patients treated with surgery in a single tertiary care center.
They found a strong relationship between different variables and the rate of long-term remission and recurrence. Some parameters have been already assessed in the literature but data are heterogeneous.
The paper is well structured and the topic is interesting because they combined many different parameters to assess the prognosis.
My main concern is about the definition of recurrence.
Comments:
- abstract:
Conclusions are missing
Other suggestions: English language revision is needed. Brackets have been incorrectly used. I think it is better not to cite references in the abstract section.
Methods
- line 100: how many patients were treated with medical therapy before surgery? this data could impact the assessment of GH/IGF1 after surgery
- line 104: you say "and or" GH suppression after OGTT. I suppose that ALL patients you considered in remission presented normal IGF-1 levels at the last evaluation, please specify better the criteria for disease remission. The guidelines you cited specify the difference between "surgical remission" (GH nadir < 0.14 ng/ml) and "control" (GH nadir < 1 ng/ml), and all patients in remission need to present normal IGF-1 levels.
- line 153, what do you mean by "all criteria for remission"? Normal IGF1 and GH nadir < 1 ng/ml ? please specify
- line 177: I suppose that IGF-1 values have been included in the statistical analyses using SDS or ULN for adequate reference ranges, please specify
Results
My main concern is the lack of definition of "disease recurrence".
This event is largely analyzed in the text but the definition is missing.
(biochemical? what parameter has been considered for recurrence? IGF-1 and or GH? radiological? both?). For example, it has been proved that patients who presented normal IGF-1 after surgery but did not normalize GH nadir can show a recurrence. How many presented abnormal GH nadir with normal IGF-1 after surgery? The rate of recurrence in this series is higher than reported in most recent studies, please explain this result (see for example Freda, Pituitary 2021, Prognostic value of nadir GH levels for long-term biochemical remission or recurrence)
The result section needs to be extensively revised to explain these data.
- 196: is it the age at diagnosis, surgery or last follow up?
- 197: a macroadenoma is ≥ to 10 mm
- 203 how do you assess the resection rate? I suppose using the volume calculation of the lesion, please specify
- Table 2 and Table 3: it is not clear the relationship between the variable and the result (for example: grade 2b is significantly related to the "lack" of long-term remission), please provide a better description in the table and the text.
Moreover, I suggest:
- 191 pay attention to using brackets
- 196 pay attention to the English language
- Table 1: it is recommended to report IGF-1 values also using SDS or ULN
Discussion
- 275-281 you discuss the role of SSTR expression which is not a parameter assessed in your study, please specify that. Moreover, despite much data about this topic being available, this assessment is not now recommended by WHO in GH-secreting tumor evaluation, please specify.
- 294 I suggest citing also pasireotide in second-line treatments
Reviewer 2 Report
Article “The prognostic-based approach in growth hormone-secreting pituitary neuroendocrine tumors (PitNET): Tertiary reference center, single senior surgeon, and long-term follow-up” by Abel Ferrés and colleagues is very interesting and important study on up to date assessment of prognostic markers in acromegaly.
It is well designed, written and clear.
However there are important methodological issues which need to be clarified and completed before publication.
Inclusion criteria:
Figure 2.
1.According to the data given in the figure 2, out of 394 patients operated 54 were GH secreting PitNETs. Ten out them (54) “without long term follow up”
have been excluded.
However, in the methodological part there is a statement: “patients with incomplete recorded clinical data, or subjects who received previous pituitary surgery, or radiotherapy were excluded”. Can you clarify how many patients and for what reasons have been excluded, if applicable please correct the figure 2.
2. Methodology. Can you please complete the data on the IGF-1 and GH laboratory methods/ assays used.
3. Methodology, Results: Table 1;
Can you please complete the data regarding the IGF-1 values.
How did you calculate the median value, is it age and sex adjusted?.
The level of IGF-1 should be expressed as a baseline value, and e.g. as percentage of IGF-1 of the upper limit of age-adjusted normal range (%ULN IGF-1). Please give the normal IGF-1 ranges for female and male patients for age groups.
Please clarify and complete this issue.
4.Results
Pit NET classification data (1-3), as one of the most powerful prognostic marker deserve more visible and detailed presentation.
What was the median Ki67?
A table with specific data ( Ki67, p27, mitoses, p53, and invasion: histological and or radiological signs of cavernous or sphenoid sinus invasion) would improve presentation on those very important prognostic markers.
